# The Orthopedic-Vascular Multidisciplinary Approach Improves Patient Safety in Surgery for Musculoskeletal Tumors: A Large-Volume Center Experience

**DOI:** 10.3390/jpm11060462

**Published:** 2021-05-24

**Authors:** Andrea Angelini, Michele Piazza, Elisa Pagliarini, Giulia Trovarelli, Andrea Spertino, Pietro Ruggieri

**Affiliations:** 1Department of Orthopedics and Orthopedic Oncology, University of Padua, Via Giustiniani 2, 35128 Padova, Italy; andrea.angelini@unipd.it (A.A.); dr.elisa.pagliarini@gmail.com (E.P.); giuliatrovarelli87@gmail.com (G.T.); 2Department of Surgery, Oncology and Gastroenterology, University of Padova, 35128 Padova, Italy; 3Department of Vascular and Endovascular Surgery, University of Padua, 35128 Padova, Italy; michele.piazza@unipd.it (M.P.); andrea.spertino@gmail.com (A.S.)

**Keywords:** limb salvage, patient safety management, vascular bypass, soft tissue sarcoma, vascular reconstruction

## Abstract

**Objective:** Wide-margin resection is mandatory for malignant bone and soft tissue tumors. However, this increases the complexity of resections, especially when vessels are involved. Patients in this high-risk clinical setting could be surgically treated using the multidisciplinary orthopedic-vascular approach. This study was carried out in this healthcare organization to evaluate patient safety in term of oncologic outcomes and reduction of the complication rate. **Materials and Methods:** We retrospectively reviewed 74 patients (37 males, 37 females; mean age 46 years, range 9–88) who underwent surgical excision for bone/soft tissue malignant tumors closely attached to vascular structures from October 2015 to February 2019. Vascular surgery consisted of isolation of at least one vessel (64 patients), bypass reconstruction (9 patients), and end-to-end anastomosis (1 patient). Mean follow-up was 27 months. Patients’ demographics, tumor characteristics, adjuvant treatments, type of orthopedic and vascular procedures, and oncologic and functional outcomes and complications were recorded. **Results:** Overall survival was 85% at 3 years follow-up. In total, 22 patients experienced at least one major complication requiring further surgery and 13 patients experienced at least one minor complication, whereas 17 reported deviations from the normal postoperative course without the need for pharmacological or interventional treatment. Major complications were higher in pelvic resections compared to limb-salvage procedures (*p* = 0.0564) and when surgical time was more than 4 h (*p* = 0.0364) at univariate analysis, whereas the most important multivariate independent predictors for major complications were pelvic resection (*p* = 0.0196) and preoperative radiotherapy (*p* = 0.0426). **Conclusions:** A multidisciplinary ortho-vascular approach for resection of malignant bone and soft tissue tumors tightly attached to important vascular structures should be considered a good clinical practice for patient safety.

## 1. Introduction

Patient care is changing over time because of the improvement of technology, pharmacology, and surgical techniques. The most important predictor of local recurrence after surgical excision of bone and soft tissue malignant tumors is negative resection margins. Before the advent of chemotherapy, the primary surgical treatment for bone tumors of the extremities was amputation, whereas today, secondary to the advances in adjuvant treatments and surgical techniques, limb-salvage surgery has been shown to be feasible with adequate margins in >90% of cases [1,2,3]. However, involvement of neurovascular bundles challenges negative resection margins. The pioneers in the field of musculoskeletal oncology anticipated the essential role of the multidisciplinary orthopedic and vascular surgery approach to treat patients with bone and soft tissue malignant tumors [4]. It is clear that the proper management of major vessels is part of the routine work of an orthopedic oncology surgeon, but this poses challenges to patient safety with a need for change in the way we approach patient care in surgery. A surgical team in which orthopedic and vascular surgeons cooperate has been associated with decreased morbidity and complications, and improved outcomes for the patients [5,6,7,8,9].

We performed a retrospective analysis of patients who underwent surgical excision for bone/soft tissue malignant tumors closely attached to vascular structures, using a multidisciplinary orthopedic and vascular surgery approach and aiming to evaluate patient safety in terms of oncologic outcomes and reduction of the complication rate.

## 2. Materials and Methods

We retrospectively studied all patients with malignant bone and soft tissue tumors that were treated using a multidisciplinary approach combining the expertise of orthopedic oncology and vascular surgeons from October 2015 to February 2019. We intentionally excluded from the analysis all patients treated after February 2019 to have a potential minimum follow-up of 2 years. From a total of 493 operations for musculoskeletal tumors, in 393 operations a vascular surgeon was not required, in 24 operations a vascular surgeon was on call but never scrubbed-in, and in 2 operations a vascular surgeon was called in emergency for intraoperative vascular complications (Figure 1).

The above exclusions left us with 74 patients (37 male and 37 female patients; mean age, 46 years; age range, 9–88 years) who underwent combined ortho-vascular surgery that required orthopedic oncology en bloc tumor resection and vascular surgery for the protection/isolation and/or reconstruction of major vessels. The mean follow-up was 27 months (range, 24–44 months). All patients or their relatives gave written informed consent for their data to be included in scientific studies. An Institutional Review Board/Ethics Committee approval was not required for our retrospective study with fully anonymized clinical routine data.

Details of patients’ age, gender, comorbidities, tumor histology, grade, staging and site, medical history, imaging studies, and oncological management, including resection and reconstruction, vascular reconstruction, additional procedures, and the need for adjuvant treatments (radiation therapy and chemotherapy), were recorded and analyzed: 33 patients had at least one cardiovascular risk factor including smoking, obesity, type 2 diabetes mellitus, arterial hypertension, and hyper-cholesterol; 1 patient had coronary artery disease; 4 patients had peripheral arterial disease; and 3 patients had a history of deep venous thrombosis; furthermore, 14 patients were classified as American Society of Anesthesiologists (ASA) physical status classification system 1, 45 patients as ASA 2, and 15 patients as ASA 3 (Table 1).

All patients underwent preoperative radiographic (for bone tumors), computed tomography (CT), and magnetic resonance (MR) imaging staging. The average tumor volume was 297 cc (median, 102 cc; range, 3–4082 cc), which was an average of 162 cc (median, 102 cc; range, 12–942 cc) for bone tumors and an average of 599 cc (median, 133 cc; range, 3–4082 cc) for soft tissue tumors. The average maximum diameter of the tumor was 8.9 cm (median, 8 cm; range, 3–30 cm), which was an average of 8.0 cm (median, 7 cm; range, 3.2–15 cm) for bone tumors and an average of 11.2 cm (median, 12 cm; range, 3–30 cm) for soft tissue tumors. The difference in volume and diameter between bone and soft tissue tumors was not statistically significant (*p* = 0.123 and *p* = 0.063, respectively). In 67 patients a needle or trocar biopsy was done preoperatively; in 7 patients, biopsy was not done because the tumor had pathognomonic characteristics on imaging (2 patients) or was an obvious local recurrence (5 patients). CT angiography was routinely performed to assess the vascular anatomy and its relation to the tumor in order to plan an adequate dissection or possible reconstruction. In patients with inconclusive CT angiography, digital subtraction angiography was performed.

Perioperative adjuvant treatments included chemotherapy in 28 patients, radiotherapy in 16 patients, combined chemotherapy and radiotherapy in 12 patients, and selective arterial embolization in 6 patients. Surgical treatments included removal of primary tumors in 57 patients and local recurrences in 17 patients. Additionally, seven patients were treated with forequarter amputation (two cases) or hindquarter amputation (five cases) as primary treatment. Reconstruction of bone defects after tumor resection was done in 51 patients with a megaprosthesis (41 patients), a custom-made 3D-printed pelvic prosthesis (7 patients), an expandable proximal tibia megaprosthesis (1 patient), a conventional revision hip prosthesis (1 patient), and a massive distal femur bone allograft (1 patient). Surgical margins were histologically defined on the basis of the worst margin on the specimen according to Enneking [10]: wide if a continuous shell of healthy tissue could be demonstrated around the tumor (53 patients; 72%), marginal if the plane of resection was along the pseudo-capsule (15 patients; 20%), and intralesional when pathological tissue was present in a margin (6 patients; 8%). Moreover, the surgical margins were also identified according to the R categories defined by the Union for International Cancer Control (UICC), with R0 representing no macroscopic or microscopic residual tumor postoperatively (68 patients; 92%), R1 microscopic (4 patients; 5%), and R2 macroscopic residual tumor (2 patients; 3%), respectively [11].

Vascular surgery during en bloc tumor resection included isolation of at least one vessel strictly related to the tumor with the possibility of preserving it in 64 patients, bypass vascular reconstruction in 9 patients, and end-to-end vascular anastomosis in 1 patient. In four patients the major artery only was reconstructed (Type II reconstruction) [12], and in five patients the major artery and vein were reconstructed (Type I reconstruction) [12]. The contralateral great saphenous vein was used for the bypass venous reconstruction in all patients, and for the arterial bypass reconstruction in eight patients (Figure 2); a polytetrafluoroethylene (PTFE) vascular graft was used for a femoro-popliteal arterial bypass in one patient because the contralateral great saphenous vein was not adequate.

After the ortho-vascular surgery, plastic surgery wound coverage was necessary in 21 patients using the medial gastrocnemius flap (13 patients) or local myocutaneous flaps (8 patients). The mean duration of the ortho-vascular surgery was 270 min (range, 65–770 min), and the mean blood loss was 770 mL (range, 50–4600 mL). As expected, the surgical time and blood loss was higher for major resections and reconstructions such as pelvic tumors resections.

Routine follow-up examinations were performed every 3 months for the first 2 years, every 6 months for the next 3 years, and then annually. Follow-up examinations included physical examination and functional evaluation, imaging studies, and disease-specific imaging. Oncologic results were evaluated with respect to local recurrence, metastasis, or death, and the patients were classified as having no evidence of disease (NED), being disease free after treatment of local recurrence (NED-LR) or metastasis (NED-M), being alive with disease because of local recurrence or metastasis (AWD), and being dead of disease (DWD). Survival was defined as the time from surgery to last follow-up or death. Complications were recorded and graded according to the Clavien–Dindo classification of surgical complications [8,9]. In summary, complications were divided in five grades: Grade (I)—any deviation from the normal postoperative course without the need for pharmacological or interventional treatment; Grade (II)—requiring pharmacological treatment with drugs; Grade (III)—requiring surgical, endoscopic, or radiological intervention; Grade (IV)—life-threatening complication requiring intermediate care (IC)/intensive care unit (ICU); Grade (V)—death [8].

Categorical variables were expressed as percentages of the total patients in a category. The mean, standard deviation, and range of all continuous variables were calculated. The effect level of clinical characteristics on outcomes was evaluated using the univariate Kaplan–Meier analysis as a time-event analysis. Comparison of the curves was done in a bivariate analysis with the log-rank test. Logistic binary regression was used for analyzing if there one or more independent variables that influence the rate of major complications (measured as a dichotomous variable). Differences were considered statistically significant when the *p* value was less than 0.05. The data were recorded in a Microsoft Excel1 2003 spreadsheet and analyzed using Med-Calc software version 11.1 (MedCalc Software, Mariakerke, Belgium).

## 3. Results

### 3.1. Oncological Outcome

Mean follow-up was 27 months. At 3 years of follow-up, the overall survival of the patients was 85% (Figure 3).

At the last follow-up, 39 patients were NED, 7 patients were NED-LR, 3 patients were NED-M, 1 patient was NED-LR/M, 17 patients were AWD, and 7 patients were DWD. The overall survival to local recurrence was 64% (Figure 4) and the overall survival to metastasis was 58% (Figure 5). We observed that patients with no evidence of disease at the last follow-up were 16% (1/6 patients) in those treated with intralesional margins, 67% (10/15 patients) in marginal margins, and 55% (29/53 patients) in wide margins.

### 3.2. Complication Rate

In total, 22 patients experienced at least one major complication (Grade III), 13 patients experienced at least one minor complication (Grade II), whereas 17 reported deviation from the normal postoperative course without the need for pharmacological or interventional treatment (Grade I) (Table 2).

No patient experienced limb ischemia during the follow up, even if in two patients, a subtotal occlusion of the venous bypass was observed at the Doppler ultrasonography that, however, did not require any further management for the patients. Deep hematoma and wound-related problems with/without infection were the most common major complications (Table 2). A deep hematoma or sieroma was observed in five patients, but revision operation in these patients showed active bleeding from the dissected tissues without any bypass leakage. Wound dehiscence was treated with surgical debridement and pedicle flaps, especially in the four cases with large wound necrosis. One patient with major complication (deep infection) underwent final amputation after several inefficient surgical debridements.

Wound dehiscence was the most common minor complication (five patients), followed by superficial infection and sieroma (two patients each) that were treated effectively conservatively with wound dressing and pharmacological treatment. Four patients experienced deep vein thrombosis (DVT) treated with drugs, but none of these patients had a vascular reconstruction. Edema of the limb was observed in six patients with vascular reconstructions at the early postoperative period and was treated successfully with compression stockings. Temporary sensory nerve deficits (paresthesia, hypoesthesia) were reported in 19 patients, and temporary motor deficits (muscles weakness and atrophy) in 12 patients. Seven amputees reported phantom limb pain for which they took analgesic therapy. The most important univariate predictors for major ortho-vascular complications were a pelvic resection compared to a limb-salvage resection (*p* = 0.0564), as well as a surgical time of more than 4 h (*p* = 0.0364) (Table 3). The most important multivariate independent predictors for major ortho-vascular complications were pelvic resection (*p* = 0.0196) and preoperative radiotherapy (*p* = 0.0426) (Table 4).

## 4. Discussion

Musculoskeletal tumors are a rare heterogeneous group of neoplasms. Appropriate management of the patients from the diagnosis and treatment to the follow-up should be done in specialized centers, which can ensure extensive experience and a multidisciplinary approach based on a team composed by orthopedic oncology surgeons, vascular surgeons, and plastic surgeons, if necessary, to aim for the best successful surgical results and adequate margins achieving acceptable outcomes [13]. This has been shown in the present study; a combined ortho-vascular approach for malignant bone and soft tissue tumor patients provided the best surgical outcome with a low rate of major local complications. The retrospective design of the study and heterogeneous group of patients are limitations with possible selection biases; however, retrospective studies are useful for the evaluation of treatment approaches. Moreover, the number of samples and the heterogeneity in diagnoses are related to the rarity of individual tumors, despite the fact that our institute is a national reference center. Because of the relatively small number of patients in some of our histologic subtypes, we could not analyze all confounding variables with a multivariate regression model; in fact, we had the choice to reduce the number of variables to increase the value of our analysis and focused the results on complications. Moreover, we did not want to run a large number of post hoc analyses to assess the influence of some variables on oncologic outcome (such as chemotherapy induced necrosis, surgical margins, etc.) that have been clearly studied before. Finally, the lack of a control group did not allow for a rigorous interpretation of the clinical significance of oncological and vascular outcomes.

Vascular surgery contribution in orthopedic oncology surgery relates to intraoperative support in tumor resection and vascular reconstructions. Preoperative ortho-vascular planning aims to study the patency of the contralateral great saphenous vein and preparation of sterile field for harvesting, if necessary, to insert temporary shunts after resection of the tumor en bloc with the vessels, which makes reconstruction of bone defects with megaprostheses easier and allows for the perfusion of the limb before vascular reconstruction, as well as to preserve the popliteal artery branch to the medial gastrocnemius muscle head in proximal tibia reconstructions when a rotating flap of the medial gastrocnemius muscle is required [13]. Awad et al. evaluated their experience including a vascular surgeon in a multidisciplinary team for treatment of 63 patients with soft tissue sarcomas [7]. A vascular surgeon was requested for bypass reconstruction (12.5%), vessel reconstruction (25%), and vessel ligation (62.5%) [7]. In studies on soft tissue tumors of the lower extremities the incidence of vascular reconstructions was 4%–9% [14,15,16,17]. Most clinical studies regarding cooperation with vascular surgeons in orthopedic oncology included only patients that required vascular reconstruction with bypass after the en bloc excision of a tumor involving vascular bundles. Fortner et al., in 1977, were the first to demonstrate the feasibility of vascular reconstruction after en bloc tumor resection [18]. Vessel en bloc excision with the tumor specimen allows for wide margins without violating the tumor capsule, while at the same time, vessel reconstruction provides for restoration of the limb’s vascularization. In that study, in a small sample size, the authors reported no case of leg ischemia or gangrene; edema was the most common complication [19]. Other studies confirmed that tumor involvement of the vascular bundle is not an absolute indication for amputation, provided that vascular bypass can be performed [14,19,20]; in these studies, the local recurrence and metastases rate of the patients treated with en bloc resection involving major vessels was similar to those of patient treated differently.

On the other hand, the role of venous reconstruction in vascular surgery for tumors is not clear. Fortner et al. recommended routine venous reconstruction to avoid edema to the limb [18]. Another study in 23 patients with bone and soft tissue sarcomas treated with en bloc resection with arterial and venous reconstruction reported a higher incidence and a longer duration of edema in the group of patients treated with arterial reconstruction only [21]. Similarly, Hohenberger et al. in 20 cases of soft tissue sarcomas treated with en bloc resection, including neuro-vascular bundle and reconstruction with bypass (arterial bypass in 9 patients and venous bypass in 11 patients), reported edema in only 2 cases; they observed that in the case of resection of the external iliac vein or the superficial femoral vein, the ability of the great saphenous vein and lymphatic vessels is adequate if not resected with the tumor specimen [19]. Other authors reported no significant difference in complications and function in a comparison study between 12 patients treated with arterial reconstruction and 13 patients with both venous and arterial reconstruction [17]. Faenza et al. observed that venous reconstruction has some advantages postoperatively, but in the long term, they observed edema developing in all their patients [22]. Therefore, when resection is extensive and involves superficial and deep veins, venous reconstruction is recommended [19,23,24,25,26]; the superficial femoral vein and the popliteal vein should not be reconstructed, especially if the great saphenous vein is preserved [27], and if the vein is occluded, there is an absolute contraindication to its reconstruction [24]. In the present study and our practice, venous bypass reconstruction was performed when a significant compromise of the venous flow was expected after resection; in two patients, subtotal occlusion of the bypass was diagnosed at follow-up, without any complications, and in one patient, the venous reconstruction was not performed because the vein was compromised.

Currently, the most used grafts for vascular bypasses are the autologous vein and the synthetic grafts (ePTFE or Dacron). Some authors suggested the use of autologous large saphenous vein bypass [28], whereas other authors did not find any differences in terms of long-term patency between synthetic prostheses and autologous vein grafts [12]. The main concern for synthetic vascular grafts is the risk of infection. Adelani et al. in 14 patients with soft tissue sarcomas treated by resection and vascular bypass reconstructions reported no superiority of the autologous vein prosthesis over the synthetic prosthesis relative to the risk of infection, even if the latter appeared to increase the risk of wound dehiscence [29]. Other studies reported that autologous vein prosthesis has a higher long-term patency rate and lower risk of infection [12,19,25,26,30,31,32,33,34]. In our practice, the contralateral great saphenous vein was used as the first choice; a synthetic graft was used only in one patient because the contralateral great saphenous vein was too short after harvesting. Synthetic vascular grafts are a valid alternative in cases where an autologous vein is not available or there is a significant discrepancy in the diameters of the vessels to be reconstructed. A further aspect to consider for the choice of synthetic vascular grafts is the anatomical site; above the knee both autologous vein and synthetic vascular grafts can be used, while below the knee autologous veins are preferable [19,27].

Vascular reconstructions in orthopedic oncology surgery do not have a negative effect on the survival of the patients. Some authors reported a significantly lower survival of patients treated with vascular reconstructions, even if a selection bias of a locally more aggressive neoplasm should be considered [15]. Poultsides et al. compared the outcomes of two groups of soft tissue sarcoma patients [14]. The first group included 50 patients undergoing resection and vascular reconstruction and the second group included 100 patients without vascular reconstructions; they reported no statistically significant differences in the 5-year overall survival between the two study groups (group I, 59%; group II, 53%; *p* = 0.067) [14]. In the present study, the overall survival was good; however, we did not include a control group for comparison analysis. Moreover, the well-known role of surgical margins for local control and overall survival on malignant tumors should be considered in oncologic outcome. In our series we included several histotypes, with sometimes challenging surgeries, which justifies the relative low incidence of consecutively NED patients (55%) with adequate margins. Different scores have been used for the evaluation of the function of the patients with musculoskeletal tumors [3,35,36]. Ghert et al. compared the function of the patients after lower limb soft tissue sarcoma surgery, with and without vascular reconstruction. They found no statistically significant difference of function between the group with vascular reconstructions (mean score, 78.5%) and the group without vascular reconstructions (mean score, 82.2%) [37]. Other authors reported similar results with respect function in sarcoma patients with resection and vascular reconstructions (mean score, 70%–80% [12,31,38,39]. In the present study, we did not evaluate the function of the patients because data on parameters of function were not available for the majority of the patients. We observed a significant number of patients with temporary sensory and motor nerve deficits that seemed not to be related to the vascular reconstructions themselves considering the self-limiting duration of the symptoms.

Complications do occur in tumor surgery as well as in vascular repair/reconstructions. However, amputation as definitive surgery is rarely required for vascular complications after tumor resection with vascular reconstructions [12,16,27,40]. Awad et al. reported a 17.7% rate of complications, mainly superficial infections (54.5%), deep infections (27.3%), seromas (9%), and local flap necrosis (9%); in their series, complications were more common in patients undergoing hip disarticulation and hemipelvectomy [7]. We concur with this report; in the present study, the rate of complications was higher in the group of patients with pelvic surgery, maybe due to the high complexity of this type of surgery. However, we did not find a significant association between vascular reconstructions and major complications, as previously reported by other authors [14,15,30,31,40]. Davis et al. observed that the wound-healing time in patients with resection and vascular reconstruction was almost twice than that of the group of patients without vascular reconstructions (88 vs. 39 days; *p* < 0.002) with a significantly higher number of revision operations for wound complications [15]. Radiotherapy is an important predictor for major complications in tumor surgery with and without vascular reconstructions [3,15,41]. We strongly recommend a combined plastic surgery approach with soft tissue reconstruction in cases where the risk of complications is high due to a wider resection area, poor coverage of megaprostheses and allografts, and previous radiotherapy [13,26].

## 5. Conclusions

Although the lack of a control group and limitations of this study prevent us from a statistical demonstration on improved overall survival, the multidisciplinary ortho-vascular approach for the surgical treatment of patients with musculoskeletal tumors tightly attached to important vascular structures should be considered a good clinical practice for patient safety. Both consultation and cooperation with vascular surgeons are paramount, not only if vascular reconstruction is planned, but in all cases of complex tumor resections close to vascular bundles that may require intraoperative vascular surgery support for possible vascular reconstruction. In this scenario, the outcome of the patients is hypothetically expected to improve without increasing the rate of vascular reconstruction-related complications, even if further, more focused studies should be performed before including this combined approach in the routine management of these patients.

## Figures and Tables

**Figure 1 jpm-11-00462-f001:**
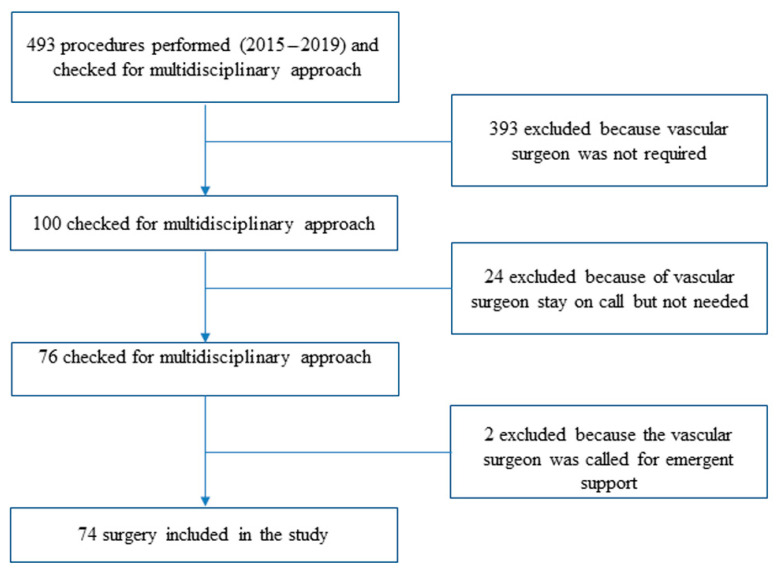
Patients’ cohort selection process.

**Figure 2 jpm-11-00462-f002:**
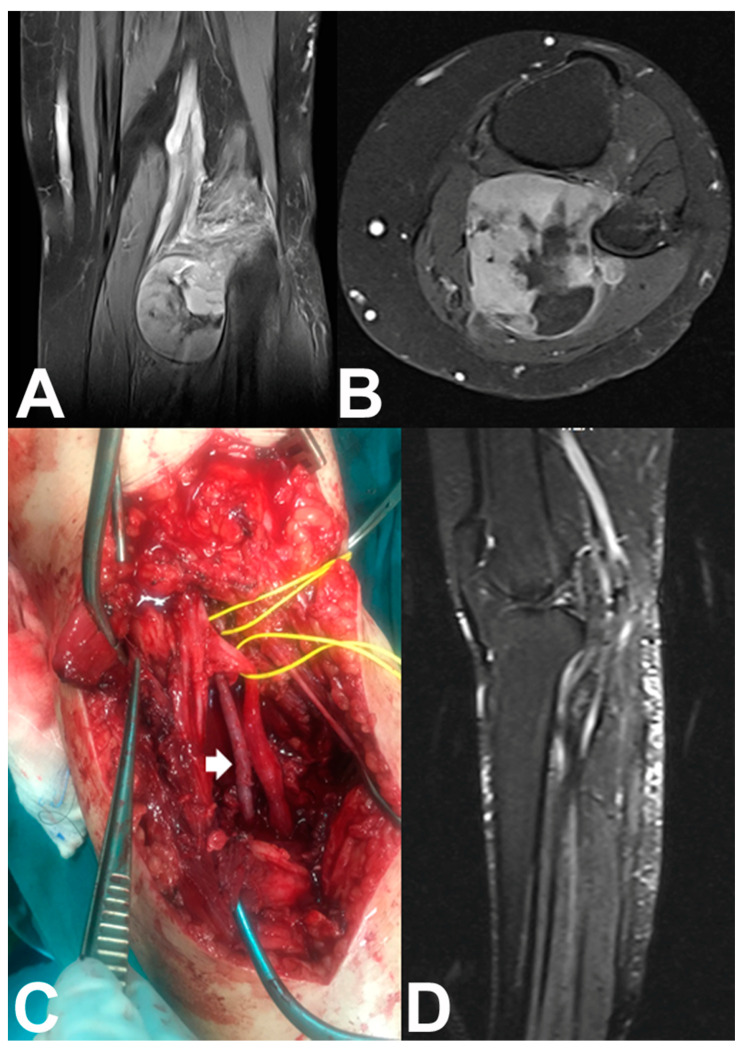
(**A**) Coronal and (**B**) axial T2-weighted MR images of the left knee of a 55-year-old woman with a popliteal fossa synovial sarcoma. (**C**) En bloc (marginal) tumor resection was done after identification and preservation of the peroneal and tibial nerves (lower vessel-loop), ligation without reconstruction of the popliteal vein, identification of the popliteal artery (upper vessel-loop), and arterial bypass reconstruction with the tibial artery using a contralateral great saphenous vein graft (white arrow) without venous bypass reconstruction. (**D**) Sagittal T2-weighted MR image shows tumor resection and limb preservation with patent anastomosis.

**Figure 3 jpm-11-00462-f003:**
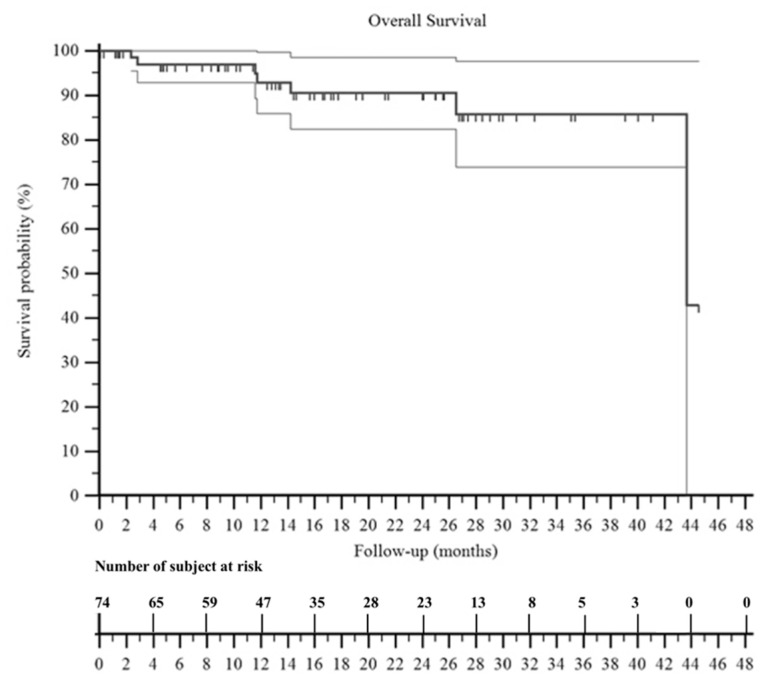
A Kaplan–Meier curve shows the overall survival of the patients included in this series. It was 92% at 2 years and 85% at 3 years. The two surrounding thin black lines represent the 95% confidence intervals.

**Figure 4 jpm-11-00462-f004:**
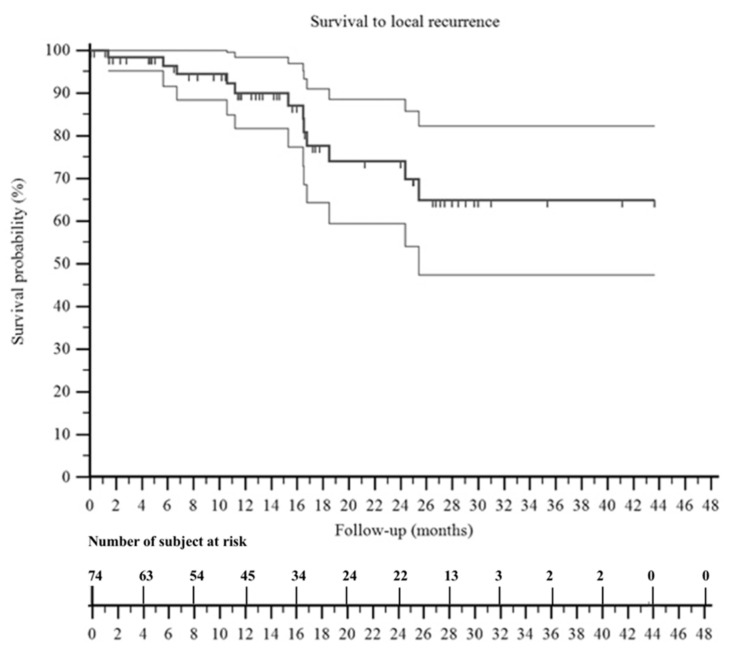
A Kaplan–Meier curve shows the survival to local recurrence of the patients included in this series. It was 74% at 2 years and 64% at 3 years. The two surrounding thin black lines represent the 95% confidence intervals.

**Figure 5 jpm-11-00462-f005:**
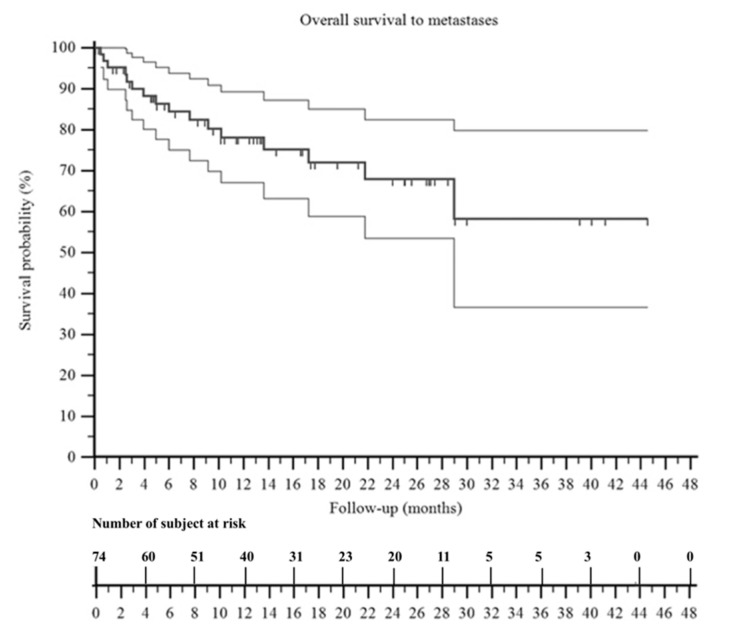
A Kaplan–Meier curve shows the survival to metastasis of the patients included in this series. It was 68% at 2 years and 58% at 3 years. The two surrounding thin black lines represent the 95% confidence intervals.

**Table 1 jpm-11-00462-t001:** Demographic details of the patients (*n* = 74) included in this series.

Data	Patients (*n*)	%
Age (mean years)	46 (range, 9–88)	-
Gender (male/female)	37/37	-
Obesity	16	21.6
Hypertension	15	20.3
Smoking	13	17.6
Dyslipidemia	6	8.1
Type II diabetes	4	5.4
Peripheral arterial disease	4	5.4
Coronary artery disease	1	1.4
Previous deep vein thrombosis	3	4.1
>1 cardiovascular risk factors	45	60.8
American Society of Anesthesiologists (ASA) Score 1	14	18.9
American Society of Anesthesiologists (ASA) Score 2	45	60.8
American Society of Anesthesiologists (ASA) Score 3	15	20.3
Bone tumors	54	73
* Symptoms:*		
Pain	38	70.4
Swelling	8	14.8
Functional limitation	9	16.7
Pathological fracture	7	13
Asymptomatic	9	16.7
* Histological diagnosis:*		
Osteosarcoma	19	35.1
Chondrosarcoma	16	29.6
Ewing’s sarcoma	3	5.6
Chordoma	1	1.9
Metastatic bone disease	11	20.4
Hematological malignancies	4	7.4
* Site:*		
Proximal tibia	15	27.8
Proximal femur	12	22.2
Pelvis/sacrum	9	16.7
Distal femur	9	16.7
Proximal humerus	6	11.1
Scapula	1	1.9
Humeral shaft	1	1.9
Proximal tibia/distal tibia	1	1.9
Soft tissue tumors	20	27
* Symptoms:*		
Mass/Swelling	13	65
Pain	8	40
Functional limitation	2	10
Asymptomatic	3	15
* Histological diagnosis:*		
Synovial sarcoma	7	35
Leiomyosarcoma	2	10
Liposarcoma	2	10
Pleomorphic sarcoma	2	10
Other	7	35
* Site:*		
Thigh	9	45
Popliteal fossa	3	15
Hip	2	10
Buttocks	2	10
Forearm	2	10
Knee	1	5
Pelvis	1	5
Metastases at time of surgery (bone and soft tissue tumors)	10	13.5
Lung metastases:	9	12.2
Skip metastases	1	1.4

**Table 2 jpm-11-00462-t002:** Complications of ortho-vascular surgery in the patients included in this series, classified according to the Clavien–Dindo system.

Data	* Postop	Early	Late	N. Events/n. pts	Relative % **	Absolute % °°
Grade I				15/17	42.50%	22.90%
Edema of the limb (13)	7	6	-			
Delayed wound healing (4)	4	-	-			
Grade II				16/13	32.50%	17.60%
Subtotal bypass occlusion (2)	-	2	-			
Superficial infection (2)	1	1	-			
Wound dehiscence and partial necrosis (5)	4	1	-			
Sieroma or haematoma (2)	-	2	-			
Deep vein thrombosis (4)	2	2	-			
Periprosthetic fracture with cast (1)	-	-	1			
Grade III				28/22	55%	29.70%
Deep hematoma or sieroma (5)	4	1	-			
Complete wound dehiscence (11)	7	3	1			
Wound necrosis and infection (4)	2	2	-			
Active bleeding (1)	1	-	-			
Deep infection (6)	2	-	4			
Prosthetic dislocation (1)	-	-	1			
Grade IV				2	5%	2.70%
Myocardial infarction (1)	1	-	-			
Systemic sepsis (1)	-	1	-			
Grade V	-	-	-	-	-%	-%

* Postoperative (<1 month from surgery), early onset (between 1 and 6 months), late (after 6 months). ** Relative percentage of subtype complication on 40 patients (that reported almost one complication). °° Absolute percentage of subtype complication on 74 patients (entire series).

**Table 3 jpm-11-00462-t003:** Risk factors for major complications (Grade III Clavien–Dindo) of ortho-vascular surgery in the patients included in this series.

Variables	Cut Off n. Events/pts Hazard Ratio (95%CI)	Cut Off n. Events/pts Hazard Ratio (95% CI)	*p*-Value
Age	<65 years	>65 years	0.9641
	17/57 (29.8%)	5/17 (29.4%)	
	HR 1.0231	HR 0.9775	
Gender	Female	Male	0.0914
	14/37 (37.8%)	8/37 (21.6%)	
	HR 2.0658	HR 0.4841	
Cardiovascular risk factors	Yes	No	0.8347
	13/45 (28.9%)	9/29 (31.0%)	
	HR 0.9124	HR 1.0960	
Type 2 diabetes mellitus	Yes	No	0.7487
	1/4 (25.0%)	21/70 (30.0%)	
	HR 1.3288	HR 0.7526	
Obesity	Yes	No	0.1726
	5/16 (31.2%)	17/58 (29.3%)	
	HR 2.0448	HR 0.8547	
Preoperative radiotherapy	Yes	No	0.1
	8/16 (50.0%)	14/58 (24.1%)	
	HR 2.3397	HR 0.4274	
Neoplasia volume	<100 mL	>100 mL	0.6754
	12/38 (31.6%)	10/36 (27.8%)	
	HR 1.1961	HR 0.8360	
Intervention time	Less 4 h	>4 h	0.0364 *
	6/35 (17.1%)	16/39 (41.0%)	
	HR 0.4083	HR 2.4491	
Vascular bypass	Yes	No	0.2772
	4/9 (44.4%)	18/65 (27.7%)	
	HR 2.1040	HR 0.4753	
Flap (yes vs. no)	Yes	No	0.2984
	8/21 (38.1%)	14/53 (26.4%)	
	HR 1.6540	HR 0.6046	
Tumor site (pelvis vs. other sites)	Pelvis	Other sites	0.0564 *
	6/10 (60.0%)	16/64 (25.0%)	
	HR 3.0753	HR 0.3252	

* Statistically significant.

**Table 4 jpm-11-00462-t004:** Logistic regression analysis to evaluate independent variables as predictors for major complications (Grade III Clavien–Dindo) in the entire series.

Variables	Odds ratio	C.I. 95%	*p*-Value
Age (<65 years)	1.4684	0.2550–8.4556	*p* = 0.6671
Gender (F)	2.3379	0.6725–8.1272	*p* = 0.1816
Cardiovascular risk factors	1.4685	0.3449–6.2517	*p* = 0.6032
Type 2 diabetes mellitus	1.4414	0.0613–33.8653	*p* = 0.8204
Obesity	2.3910	0.1913–29.8893	*p* = 0.4988
Preoperative radiotherapy	4.7287	1.0535–21.2256	*p* = 0.0426 *
Tumor volume > 100 mL	1.2882	0.3609–4.5978	*p* = 0.6965
Surgical time > 4 h	2.0073	0.4204–9.5837	*p* = 0.3823
Vascular bypass	1.8550	0.2875–11.9707	*p* = 0.5160
Flap reconstruction	3.2670	0.7090–15.0548	*p* = 0.1288
Site (pelvis)	10.6054	1.4601–77.0316	*p* = 0.0196 *

* statistically significant.

## Data Availability

Data and material are available on request.

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
