# Peer review of "The Orthopedic-Vascular Multidisciplinary Approach Improves Patient Safety in Surgery for Musculoskeletal Tumors: A Large-Volume Center Experience"

_jpm, 2021, doi:10.3390/jpm11060462_

Round 1

Reviewer 1 Report

This is a patient series of a heterogenous group of patients with primary and recurrent bone and soft tissue tumors undergoing orthopedic-vascular multidisciplinary operative treatment. The article emphasizes the multidisciplinary approach to musculoskeletal malignancies, while the novelty of findings remains debatable. I have a couple points to raise:

Methods

You state that patient with follow-up shorter than 24 months and without full postoperative data were excluded, yet it appears no patients were excluded due to these reasons. If that is correct, please clarify this in the methods section, otherwise add the information.

Results

Table 1. Please add percentage to the n of patient for each characteristic.

The average tumor volume is given separately for soft tissue sarcoms and bone sarcomas. As a number of bone tumors were not sarcomas but metastases or hematological disease and I suppose those were included in the tumor volume average, the terms soft tissue tumors and bone tumors appear more accurate.

Please specify the resection margins regarding width of healthy tissue in wide and marginal resections and give additional information on R-classification.

Kaplan Meier curves need to be improved. What do the additional lines stand for, confidence interval? Please clarify in diagram or legend.

What is your definition of major versus minor complications? Using a standardized classification such as the clavien-dindo, any complication requiring surgical intervention should be regarded major. Regarding the wound necrosis/superficial infection, these are categorized as minor complications but in the text you describe these were treated by surgical debridement and flap coverage which rather justify classifying these as major complications. If so, the risk factor analyses need to be adjusted.

It is not clear what method was used for risk factor analysis for major complications. In the methods section Kaplan-Meier and Log-rank analysis are mentioned and these represent time-event analysis. If so, what was the time cut-off? I would recommend performing logistic binary regression rather than time-event analysis to identify risk factors for complications as the occurrence rather than the timing is of relevance.

Reviewer 2 Report

Summary :

This retrospective study analyzes a sarcoma cohort of patients with a multidisciplinary surgical approach involving orthopedic and vascular surgeons.
It focuses on survival analysis, complications description, and vascular acts associated with sarcoma surgeries.

This cohort does not identify a worse survival prognosis in patients with a vascular-orthopedic approach compared to other similar studies.

Broad comments :

It is an exciting subject, as it focuses on a crucial point concerning vascular structure management in sarcoma surgery in a well-known national reference center. 
It brings an extensive cohort analysis with a report of vascular acts, related complications, and survival analysis.

Its design nonetheless must limit discussion and conclusion, as it lacks a control group. 
It seems challenging to affirm low complication rates, with almost 50% complications (major and minor), in this specific non controlled design.
Indeed, it exposes potential biases, as these patients may have more advanced conditions. The margin quality rates might confirm this hypothesis and raise the question of more advanced lesions as well. The elevated marginal resection rate might be interesting to discuss more profoundly and link it to survival.
The discussion might focus on study objectives (vascular surgeries associated, survival, complications) and should aim to follow question driven paper format.

Specific comments :

In table 2 l.168: 
one amputation is reported, in l.185: "7 amputees" is mentioned. This might be clarified.

In table 4 l.195:

The second column does not show OR, but copy the first column
The multivariate analysis model is not fully described; maybe its global significance should be an interesting adjunction. It might be interesting to discuss why the pelvic site disappears in the multivariate analysis.

Reviewer 3 Report

The aim of your study is to assess the value of a close peri- and intraoperative collaboration of vascular and orthopedic surgeons for the resection of soft tissue and bone sarcoma with affection of vascular structures. Although I personally do believe in the importance and value of this approach, I have substantial doubts that this can be shown with the present study design and available data. As you realize yourself, your study lacks a control group. Therefore, any conclusion stating that the orthopedic-vascular surgery collaboration improves the oncologic outcome of patients is not based on evidence but merely hypothetical. Moreover, you base your conclusion on overall survival calculated for the overall study population. However, this population is very heterogeneous with regard to tumor biology - both bone and soft tissue sarcoma, and many different histological subtypes are included - and probably also to tumor stage and grade. Moreover, co-variables which you fail to control for such as adjuvant chemotherapy possibly have a substantial effect on survival. Lastly, survival of your study population is also presented in an unconventional and in my view wrongful way, which makes interpretation and comparison of your single group's data very difficult. I assume these limitations of your study can't be changed, but you should at least address them more clearly in your discussion and phrase your conclusions more cautiously.

I have a number of other points which require improvement:

  • Please provide exact Information on how many patients were excluded from the study due to incomplete follow-up.
  • In table 1, it would be helpful to provide proportions/percentages.
  • Survival times are presented in a quite unusual way. If I understand it correctly, you give the proportion of patients alive (or recurrence-free) at the end of their respective follow-up? This is obviously a measure which is very hard to interpret, because follow-up times vary greatly between patients. I suggest either providing mean or median survival or survival rates at a fixed time interval, e.g. two years. Moreover, you need to provide mean follow-up for your study population also in the main text and not only the abstract, and the numbers of patients at risk at the different time points in the Kaplan-Meier curves.
  • Which definition or classification did you use for postoperative complications? How were major and minor complications exactly defined?
  • In table 2, you show that two patients had bypass occlusion, but in the text you write that no patient suffered from leg ischemia during follow-up. This appears contradictory. Or was the bypass effectively still patent (in the text, you then talk of "subtotal" occlusion, what does that mean?). Or does this relate to venous bypasses, which then needs to be made clear?
  • The sentence "However, these deficits should be attributed to the extent of the resections and not to the vascular surgery." is misplaced in the results sections, as it is a critical appraisal of a result. It could be moved to the discussion section, but then you should also justify why you think this is the case. To me, this sounds like mere speculation.
  • I assume that in tables 3 and 4 and the associated text, "radiotherapy" refers to "preoperative radiotherapy". I would write this explicitly, as otherwise this is prone to misunderstanding.

Author Response

The orthopaedic-vascular multidisciplinary approach improves patient safety in surgery for musculoskeletal tumors:

a large volume center experience

Reviewer Remarks

Authors’ Responses

Reviewer 1:

(x) Moderate English changes required

Introduction (Yes)

Research design (Can be improved)

Methods (Must be improved)

Results (Must be improved)

Conclusions (Yes)

Thank you Reviewer 1. As suggested, the manuscript has been reviewed by a native English speaker and corrections have been highlighted in red.

Reviewer 1.

This is a patient series of a heterogenous group of patients with primary and recurrent bone and soft tissue tumors undergoing orthopedic-vascular multidisciplinary operative treatment. The article emphasizes the multidisciplinary approach to musculoskeletal malignancies, while the novelty of findings remains debatable. I have a couple points to raise

Thank you Reviewer 1. The manuscript has been modified according to your and Reviewers’ comments.

Methods

You state that patient with follow-up shorter than 24 months and without full postoperative data were excluded, yet it appears no patients were excluded due to these reasons. If that is correct, please clarify this in the methods section, otherwise add the information.

Thank you Reviewer 1 for your comment. That sentence has been removed because incorrect. We modify the text as follow:

“We retrospectively studied all patients with malignant bone and soft tissue tumors that were treated in a multidisciplinary approach combining the expertise of orthopaedic oncology and vascular surgeons from October 2015 to February 2019. We intentionally excluded from the analysis all patients treated after February 2019 to have a potential minimum follow-up of 2 years. From a total of 493 operations for musculoskeletal tumors, in 393 operations a vascular surgeon was not required, in 24 operations a vascular surgeon was on call but never scrubbed-in, and in 2 operations a vascular surgeon was called in emergency for intraoperative vascular complications (Figure 1).

Table 1. Please add percentage to the n of patient for each characteristic.

Thank you Reviewer 1. The table as been updated with percentages for each characteristic

The average tumor volume is given separately for soft tissue sarcomas and bone sarcomas. As a number of bone tumors were not sarcomas but metastases or hematological disease and I suppose those were included in the tumor volume average, the terms soft tissue tumors and bone tumors appear more accurate.

Thank you Reviewer 1. We changed the text according with your prompt observation.

“The average tumor volume was 297 cc (median, 102 cc; range, 3-4082 cc) that was an average of 162 cc (median, 102 cc; range, 12-942 cc) for bone tumors and an average of 599 cc (median, 133 cc; range, 3-4082 cc) for soft tissue tumors”.

Please specify the resection margins regarding width of healthy tissue in wide and marginal resections and give additional information on R-classification.

Thank you Reviewer 1. We changed the text according to your suggestion:

The surgical margins were wide in 53 patients, marginal in 15 patients, and intralesional in six patients.

“Surgical margins were histologically defined on the basis of the worst margin on the specimen according to Enneking [cit]: wide if a continuous shell of healthy tissue could be demonstrated around the tumor (53 patients; 72%); marginal if the plane of resection was along the pseudocapsule (15 patients; 20%); intralesional when pathological tissue was present in a margin (6 patients; 8%). Moreover, the surgical margins were also identified according to the R categories defined by the Union for International Cancer Control (UICC), with R0 representing no macroscopic or microscopic residual tumor postoperatively (68 patients; 92%), R1 microscopic (4 patients; 5%) and R2 macroscopic residual tumor (2 patients; 3%), respectively [cit]”.

Enneking WF: A system of staging musculoskeletal neoplasms.Clin Orthop Relat Res 1986;204:9–24.

Wittekind C, Compton C, Quirke P, et al.: A uniform residual tumor (R) classification: Integration of the R classification and the circumferential margin status. Cancer 2009;115:3483–3488.

Kaplan Meier curves need to be improved. What do the additional lines stand for, confidence interval? Please clarify in diagram or legend.

Thank you Reviewer 1. We clarify in the legend.

Figure 3. A Kaplan-Meier curve shows the overall survival of the patients included in this series. It was 92% at 2 years and 85% at 3 years. The two surrounding thin black lines represent the 95% confidence intervals.

Figure 4. A Kaplan-Meier curve shows the survival to local recurrence of the patients included in this series. It was 74% at 2 years and 64% at 3 years. The two surrounding thin black lines represent the 95% confidence intervals.

Figure 5. A Kaplan-Meier curve shows the survival to metastasis of the patients included in this series. It was 68% at 2 years and 58% at 3 years. The two surrounding thin black lines represent the 95% confidence intervals.

What is your definition of major versus minor complications? Using a standardized classification such as the clavien-dindo, any complication requiring surgical intervention should be regarded major. Regarding the wound necrosis/superficial infection, these are categorized as minor complications but in the text you describe these were treated by surgical debridement and flap coverage which rather justify classifying these as major complications. If so, the risk factor analyses need to be adjusted.

Thank you Reviewer 1. This aspect required a major revision of the manuscript, because some surgically treated complications were included erroneously into the minor category. We re-checked all data. Tables and statistical analysis have been repeated and modified according to the Clavien-Dindo classification.

Text have been extensively changed according to correct analysis

Complications were recorded and graded according to the Clavien-Dindo classification of surgical complications [8-9]. In summary, complications were divided in five grades: Grade (I)—Any deviation from the normal postoperative course without the need for pharmacological or interventional treatment; Grade (II)—Requiring pharmacological treatment with drugs; Grade (III)—Requiring surgical, endoscopic or radiological intervention; Grade (IV)—Life-threatening complication requiring intermediate care (IC)/intensive care unit (ICU); Grade (V)—death [8].

Results

“Twenty-two patients experienced at least one major complication (Grade III), 13 patients experienced at least one minor complication (Grade II) whereas 17 reported deviation from the normal postoperative course without the need for pharmacological or interventional treatment (Grade I)”.

Table 2, table 3 and table 4 were modified

It is not clear what method was used for risk factor analysis for major complications. In the methods section Kaplan-Meier and Log-rank analysis are mentioned and these represent time-event analysis. If so, what was the time cut-off? I would recommend performing logistic binary regression rather than time-event analysis to identify risk factors for complications as the occurrence rather than the timing is of relevance.

Thank you Reviewer 1. We considered your suggestion and have modified the multivariate analysis as logistic binary regression that appears more accurate.

The text, table 4 and results have been changed accordingly.

“The effect level of clinical characteristics on outcomes was evaluated using the univariate Kaplan–Meier analysis as time-event analysis. Comparison of the curves was done in a bivariate analysis with the log-rank test. Logistic binary regression was used for analyzing if there one or more independent variables that influence rate of major complications (measured as a dichotomous variable).

Reviewer 2:

(x) English language and style are fine/minor spell check required

Introduction (Yes)

Research design (Can be improved)

Methods (Can be improved)

Results (Yes)

Conclusions (Must be improved)

Thank you Editor. The manuscript has been modified according to your and Reviewers’ comments

This retrospective study analyzes a sarcoma cohort of patients with a multidisciplinary surgical approach involving orthopedic and vascular surgeons. It focuses on survival analysis, complications description, and vascular acts associated with sarcoma surgeries.

This cohort does not identify a worse survival prognosis in patients with a vascular-orthopedic approach compared to other similar studies. It is an exciting subject, as it focuses on a crucial point concerning vascular structure management in sarcoma surgery in a well-known national reference center.

It brings an extensive cohort analysis with a report of vascular acts, related complications, and survival analysis.

Thank you for your comment Reviewer 2.

Its design nonetheless must limit discussion and conclusion, as it lacks a control group.

Thank you Reviewer 2. As we reported in limitations of the study, “the lack of a control group did not allow a rigorous interpretation of the clinical significance of oncological and vascular outcomes”

It seems challenging to affirm low complication rates, with almost 50% complications (major and minor), in this specific non-controlled design.

Thank you Reviewer 2. We agree with your comment. We modify the sentence considering that the incidence of major local complications (grade III Clavien-Dindo) was 29.7%

“These have been shown in the present study; a combined ortho-vascular approach for malignant bone and soft tissue tumors patients provided for the best surgical outcome with a low rate of major local complications.”

Indeed, it exposes potential biases, as these patients may have more advanced conditions. The margin quality rates might confirm this hypothesis and raise the question of more advanced lesions as well. The elevated marginal resection rate might be interesting to discuss more profoundly and link it to survival.

Thank you Reviewer 2. We added the analysis of survival in relation with resection margins as suggested.

Results

“We observed that patients with no evidence of disease at last follow-up were 16% (1/6 patients) in those treated with intralesional margins, 67% (10/15 patients) in marginal margins and 55% (29/53 patients) in wide margins”.

Discussion

“Moreover, the well-known role of surgical margins for local control and overall survival on malignant tumors should be considered in oncologic outcome. In our series we included several histotypes, with sometimes challenging surgeries which justifies the relative low incidence of consecutively NED patients (55%) with adequate margins”.

The discussion might focus on study objectives (vascular surgeries associated, survival, complications) and should aim to follow question driven paper format.

Thank you Reviewer 2. The text have been changed according to your suggestions.

In table 2 l.168:

one amputation is reported, in l.185: "7 amputees" is mentioned. This might be clarified.

Thank you Reviewer 2. We agree that it was unclear. One amputation has been performed as final treatment of a deep infection. Seven patients were treated with major amputation as oncologic surgical treatment.

“One patient with major complication (deep infection) underwent final amputation after several inefficient surgical debridements”.

“Surgical treatments included removal of primary tumors in 57 patients and local recur-rences in 17 patients. Seven patients were treated with forequarter amputation (2 cases) or hindquarter amputation (5 cases) as primary treatment”.

In table 4 l.195:

The second column does not show OR, but copy the first column

The multivariate analysis model is not fully described; maybe its global significance should be an interesting adjunction. It might be interesting to discuss why the pelvic site disappears in the multivariate analysis.

Thank you Reviewer 2. Table 4 has been completely modified according to reviewer’s comments. The multivariate analysis has been repeated after a major revision on complications in according to Clavien-Dindo classification.

As suggested we performed a logistic regression for analyzing if there were one or more independent variables that determine an outcome.

All variables in the univariate analysis have been included (as well as tumor site)

Reviewer 3:

(x) English language and style are fine/minor spell check required

Introduction (Yes)

Research design (Can be improved)

Methods (Can be improved)

Results (Must be improved)

Conclusions (Must be improved)

Thank you Editor. The manuscript has been modified according to your and Reviewers’ comments

The aim of your study is to assess the value of a close peri- and intraoperative collaboration of vascular and orthopedic surgeons for the resection of soft tissue and bone sarcoma with affection of vascular structures. Although I personally do believe in the importance and value of this approach, I have substantial doubts that this can be shown with the present study design and available data. As you realize yourself, your study lacks a control group. Therefore, any conclusion stating that the orthopedic-vascular surgery collaboration improves the oncologic outcome of patients is not based on evidence but merely hypothetical.

Thank you Reviewer 3. We agree with your comment. The aim of this study was based on the common experience (our and of numerous colleagues) that a close collaboration of vascular and orthopedic surgeons would influence outcome in musculoskeletal oncology. However, it is difficult to realize an adequate study design considering the rarity of disease and the need of control numerous confounding factors.

We smoothed out the conclusions based on the results of the study.

“Although the lacks of a control group and limitations of the study prevent us from a statistical demonstration on improved overall survival, the multidisciplinary ortho-vascular approach for the surgical treatment of patients with musculoskeletal tumors tightly attached to important vascular structures should be considered a good clinical practice for patient safety. A consultation and cooperation with vascular surgeons are paramount not only if vascular reconstruction is planned, but in all cases of complex tumor resections close to vascular bundles that may require intraoperative vascular surgery support for possible vascular reconstruction. In this scenario, the outcome of the patients is hypothetical expected to improve without increasing the rate of vascular reconstruction-related complications, even if focused further studies should be performed before including this combined approach in the routine management of these patients”.

Conclusions of the abstract have been revised

Moreover, you base your conclusion on overall survival calculated for the overall study population. However, this population is very heterogeneous with regard to tumor biology - both bone and soft tissue sarcoma, and many different histological subtypes are included - and probably also to tumor stage and grade.

Moreover, co-variables which you fail to control for such as adjuvant chemotherapy possibly have a substantial effect on survival. Lastly, survival of your study population is also presented in an unconventional and in my view wrongful way, which makes interpretation and comparison of your single group's data very difficult. I assume these limitations of your study can't be changed, but you should at least address them more clearly in your discussion and phrase your conclusions more cautiously

Thank you Reviewer 3. We agree again with your comment.

We underlined this aspect in the limitations of the study

“the number of samples and the heterogeneity in diagnoses are related to the rarity of individual tumors, despite the fact that our Institute is a national reference center.”

However, the rarity of each tumor histotype prevents us the possibility of an homogeneous series with sufficient cases for a statistical analysis. All studies in literature about vascular reconstructions in bone and soft tissue tumors merged together the different subtypes. We better explained these aspects in the limitation paragraph and modified cautiously our conclusions.

“The retrospective design of the study and heterogeneous group of patients are limitations, with possible selection biases; however, retrospective studies are useful for the evaluation of treatment approaches. Moreover, the number of samples and the heterogeneity in diagnoses are related to the rarity of individual tumors, despite the fact that our Institute is a national reference center. Because of the relatively small number of patients in some of our histologic subtypes, we could not analyze all confounding variables with a multivariate regression model; in fact, we had the choice to reduce the number of variables to increase the value of our analysis and focused the results on complications. Moreover, we did not want to run a large number of post hoc analyses to assess the influence of some variables on oncologic outcome (such as chemotherapy induced necrosis, surgical mar-gins, etc) that have been clearly studied before. Finally, the lack of a control group did not allow a rigorous interpretation of the clinical significance of oncological and vascular outcomes.

Please provide exact Information on how many patients were excluded from the study due to incomplete follow-up.

Thank you Reviewer 3 for your comment. Text was changed due to incorrect sentences. We modify the text as follow:

“We retrospectively studied all patients with malignant bone and soft tissue tumors that were treated in a multidisciplinary approach combining the expertise of orthopaedic oncology and vascular surgeons from October 2015 to February 2019. We intentionally excluded from the analysis all patients treated after February 2019 to have a potential minimum follow-up of 2 years. From a total of 493 operations for musculoskeletal tumors, in 393 operations a vascular surgeon was not required, in 24 operations a vascular surgeon was on call but never scrubbed-in, and in 2 operations a vascular surgeon was called in emergency for intraoperative vascular complications (Figure 1).

In table 1, it would be helpful to provide proportions/percentages.

Thank you Reviewer 3. As suggested also by reviewer 1, the table has been updated with percentages for each characteristic

Survival times are presented in a quite unusual way. If I understand it correctly, you give the proportion of patients alive (or recurrence-free) at the end of their respective follow-up? This is obviously a measure which is very hard to interpret, because follow-up times vary greatly between patients. I suggest either providing mean or median survival or survival rates at a fixed time interval, e.g. two years.

Moreover, you need to provide mean follow-up for your study population also in the main text and not only the abstract, and the numbers of patients at risk at the different time points in the Kaplan-Meier curves.

Thank you Reviewer 3. We reported the oncologic outcome at final follow-up as patients with no evidence of disease (NED), disease free after treatment of local recurrence (NED-LR) or metastasis (NED-M), alive with disease because of local recurrence or metastasis (AWD), and dead of disease (DWD). This is a worldwide accepted methods for comparison of oncologic outcome in musculoskeletal oncology.

We added a specific survival rates at 2 and 3 years follow-up in the figure legend. Moreover, the Kaplan-Meier curves have been modified with numbers of patient at risk at the different time points

Which definition or classification did you use for postoperative complications? How were major and minor complications exactly defined?

Thank you Reviewer 3. This aspect required a major revision of the manuscript, because some surgically treated complications were included erroneously into the minor category. We re-checked all data. Tables and statistical analysis have been repeated and modified according to the Clavien-Dindo classification.

Text have been extensively changed according to correct analysis

“Complications were recorded and graded according to the Clavien-Dindo classification of surgical complications [8-9]. In summary, complications were divided in five grades: Grade (I)—Any deviation from the normal postoperative course without the need for pharmacological or interventional treatment; Grade (II)—Requiring pharmacological treatment with drugs; Grade (III)—Requiring surgical, endoscopic or radiological intervention; Grade (IV)—Life-threatening complication requiring intermediate care (IC)/intensive care unit (ICU); Grade (V)—death [8]”.

In the statistical analysis we focused our attention on major complications that required further surgical treatments (Grade III).

In table 2, you show that two patients had bypass occlusion, but in the text you write that no patient suffered from leg ischemia during follow-up. This appears contradictory. Or was the bypass effectively still patent (in the text, you then talk of "subtotal" occlusion, what does that mean?). Or does this relate to venous bypasses, which then needs to be made clear?

Thank you Reviewer 3. Subtotal venous bypass occlusion has been observed in two patients. In the new table 2 these patients were included as minor complications (in grade II Clavien-Dindo)

“No patient experienced limb ischemia during the follow up, even if in two patients, a subtotal occlusion of the venous bypass was observed at the Doppler ultrasonography that however did not require any further management for the patients”.

The sentence "However, these deficits should be attributed to the extent of the resections and not to the vascular surgery." is misplaced in the results sections, as it is a critical appraisal of a result. It could be moved to the discussion section, but then you should also justify why you think this is the case. To me, this sounds like mere speculation.

Thank you Reviewer 3. Maybe you are right in considering this sentence as an hypothesis of the Authors, not based on scientific data. We removed from the text.

Discussion

“We observed a significant number of patients with temporary sensory and motor nerve deficits that seems not to be related to the vascular reconstructions themselves considering the self-limiting duration of symptoms”.

I assume that in tables 3 and 4 and the associated text, "radiotherapy" refers to "preoperative radiotherapy". I would write this explicitly, as otherwise this is prone to misunderstanding.

Thank you Reviewer 3. We clarified in the text and tables.

Round 2

Reviewer 1 Report

The authors have greatly improved their manuscript and addressed all raised points sufficiently.

Reviewer 3 Report

Thank you for addressing the points I had raised in my review.